# Combining Digital Image Correlation and Acoustic Emission to Characterize the Flexural Behavior of Flax Biocomposites

**Mohamed Habibi *** and Luc Laperrière

Department of Mechanical Engineering, Université du Québec à Trois-Rivières, 3351 Bd des Forges, Trois-Rivières, QC G8Z 4M3, Canada
* Correspondence: mohamed.habibi@uqtr.ca

**Abstract:** Understanding the effect of staking sequences and identifying the damage occurring within a structure using a structural health monitoring system are the keys to an efficient design of composite-based parts. In this research, a combination of digital image correlation (DIC) and acoustic emission (AE) is used to locate and classify the type of damage depending on the stacking sequence of the laminate during flexural loading. As a first step, the results of the strain fields for unidirectional, cross-ply, and quasi-isotropic laminates were compared to discuss their global behavior and to correlate the different damage patterns with the possible failure mechanisms. The damage was then addressed using a comprehensive interpretation of the acoustic emission signatures and the K-means classification of the acoustic events. The development of each damage mechanism was correlated to the applied load and expressed as a function of the loading rate to highlight the effect of the stacking sequence. Finally, the results of DIC and AE were combined to improve the reliability of the damage investigation without limiting the failure mechanism to matrix cracking, interfacial failure, and fiber breakage, as expected by the unsupervised event clustering.

**Keywords:** flax; flexural behavior; damage behavior; acoustic emission; digital image correlation

## 1. Introduction

The transportation industry constantly develops lighter, more resistant materials to improve system reliability and efficiency [1,2]. This demand is met by composite materials, especially fiber-reinforced polymeric composites owing to their high strength, stiffness, and low density [1]. Natural fiber-reinforced composite (NFRC) materials have attracted significant attention from industry and researchers due to their minimal environmental effect and low cost [1–3]. From a manufacturing point of view, these fibers are low-cost, non-abrasive to processing equipment, cause no irritation and present less health risk, and consume less energy during fiber preparation and composite molding [4–6].

The increase in their use in automotive applications reflects the growing interest in natural fiber composites [6–9]. Natural fibers can be used as an alternative to synthetic fibers for door panels, package trays, hat racks, instrument panels, internal engine covers, sun visors, boot liners, and more structural applications such as seat backs and exterior underfloor paneling.

Composite structures are exposed to bending during their service life, where the damage of various types is progressive and affects their strength [10,11]. The multiple sources of accumulated damage with increasing bending stress induce matrix cracking, interfacial failure, and delamination at the interface between adjacent ply [2,10,11]. It is, therefore, crucial to test under such more representative loading conditions, even if accidental events occur during the servicing of automotive components. In this case, bending testing monitored with acoustic emission can be performed to obtain information on the damage type and its evolution [2,11].

Even though the acoustic emission (AE) method has been widely used to identify damage modes in laminated composites, it might not be sufficient on its own due to the

complex mechanical behavior of composites. Maillet et al. [12] showed the limitations of relying solely on AE for damage identification by investigating the effects of energy attenuation on AE signal features. Therefore, it is crucial to use additional measurement techniques alongside AE to distinguish damage modes accurately. Digital image correlation (DIC) is a technique that is widely used for strain monitoring and damage detection [13–15]. For example, Jebri et al. [16] used the DIC method to investigate the damage mechanism of notched and unnotched carbon/PPS composite plates. Pierron et al. [17] compared the damage mechanisms of open-hole composite laminates at the ply and sublaminate levels by monitoring strains at the surface of the specimens. Lomov et al. [18] used DIC to determine damage onset for textile composites. In some studies [19–21], AE and DIC methods have been used simultaneously to capture the damage modes more realistically. Oz et al. [22] pointed out that AE alone may not be sufficient for damage mode identification and suggested using surface strain mapping by DIC alongside AE. Suarez et al. [23] investigated the influence of embedded optical fiber placed at various orientations in unidirectional CFRP composites using AE and DIC simultaneously. The AE and DIC results were consistent with one another. Oz et al. [24] used AE and DIC to investigate whether high-frequency AE events always originated from fiber failure in quasi-isotropic CFRP laminates. They concluded that the AE registration with high frequency and low amplitude represents matrix cracks. Habibi and Laperrière [25] used DIC and AE methods to assess the damage mechanism of flax-based laminates with and without an open hole. They found that the contribution of different damage types to the total damage of the laminates can be characterized using the AE and DIC methods.

Innovation in composite materials lies in understanding how different parameters affect their behavior. One crucial parameter is the stacking sequence, which can have a significant impact on the composite's bending behavior. To investigate this, we conducted research to explore how lay-up sequences affect laminate behaviors. Using three common lay-up sequences ($[0]_{16}$, $[0-90]_8$, and $[0 +45 \ 90 -45]_4$), we manufactured and tested a series of composites. To track damage evolution during composite loading, we utilized digital image correlation (DIC) and acoustic emission (AE) techniques. This innovative research sheds light on the role of lay-up parameters in composite behavior, which will help the design and manufacturing of composites move forward in the future.

## 2. Materials and Methods

The samples were fabricated from Lingrove prepreg (Ekoatape P-UD 3.2). The prepreg was made of unidirectional flax fiber pre-impregnated with biobased CORAL resins (220 $g/m^2$ with 50% bio-epoxy). The composite plates were manufactured by hand lay-up of 16 prepreg layers, 400 mm wide by 400 mm long, with different stacking sequences: $[0]_{16}$, $[0 \ 90]_{8,}$ and $[0 +45 \ 90 -45]_4$ laminates, with a total thickness of 3 mm and resulting fiber content of 40% $\pm$ 0.2.

The recommended curing cycle was used for these laminates. A heat ramp of 1–2 °C/min with a dwell at 80 °C for 30 min and an additional dwell at the activation temperature of the CORAL resins (120 °C) for 30 min, followed by a cooling rate of 2–4 °C/min, was used in a press with 15 bars of pressure.

Test coupons were cut using abrasive water-jet machines according to the ASTM D790 standard for flexural tests. At least five samples for each stacking sequence were tested. Experiments were carried out using a 100 kN MTS universal testing machine and a crosshead speed of 2 mm/min.

To monitor the damage progression in laminates, DIC was used for full-field surface strain measurement. Two Imager M-lite digital cameras with a CCD matrix of five million pixels were used to record the deformation state by imaging a speckle pattern created on the sample surface using black and white spray paints. During the test, all of samples were imaged at a constant imaging frequency of 14 Hz. LaVision software was used to determine the strain states based on the deformation of the speckle pattern.

During laminate testing, a two-channel AE system supplied by Vallen with a sampling rate of 5 MHz was used for damage monitoring. The AMSY-6 software was used to record the AE events from two KRNm300 sensors. High-vacuum silicon grease was used as a coupling agent between the sensors and the surface. Load, strain, and head displacement data were fed to the Vallen system and recorded at a 10 Hz sampling frequency. A pencil lead break test was used to calibrate the data acquisition system and ensure good conductivity between the sample surface and the sensors.

## 3. Results and Discussions

### 3.1. Flexural Properties

To understand how the stacking sequence affects the laminate bending behaviors, $[0–90]_8$ and $[0 +45\ 90\ −45]_4$ laminates are loaded in two different configurations. First, both of them are loaded with the 0° layer as the bottom layer (for convenience, the layer orientation angle of the bottom layer will be underlined, in this case $[\underline{0}/90]_8$ and $[\underline{0}/+45/90/−45]_4$). In the second configuration, ±45° and 90° oriented layers are used as bottom layers ($[0/\underline{90}]_8$ and $[0/+45/90/\underline{−45}]_4$).

The left side of Figure 1 illustrates the different test configurations, while the right side shows typical stress–strain curves obtained from flexural tests. Table 1 summarizes the flexural strength and modulus calculated using the failure load, sample dimensions, and test span. As can be seen, the highest flexural modulus and strength were found in unidirectional flax composites ($[0]_{16}$), which were approximately 26 GPa and 244 MPa, respectively.

The bottom layer angle shows a significant effect on the resulting flexural behaviors. The highest properties were observed when the bottom layers were oriented at 0° ($[\underline{0}/90]_8$ and $[\underline{0}/+45/90/−45]_4$).

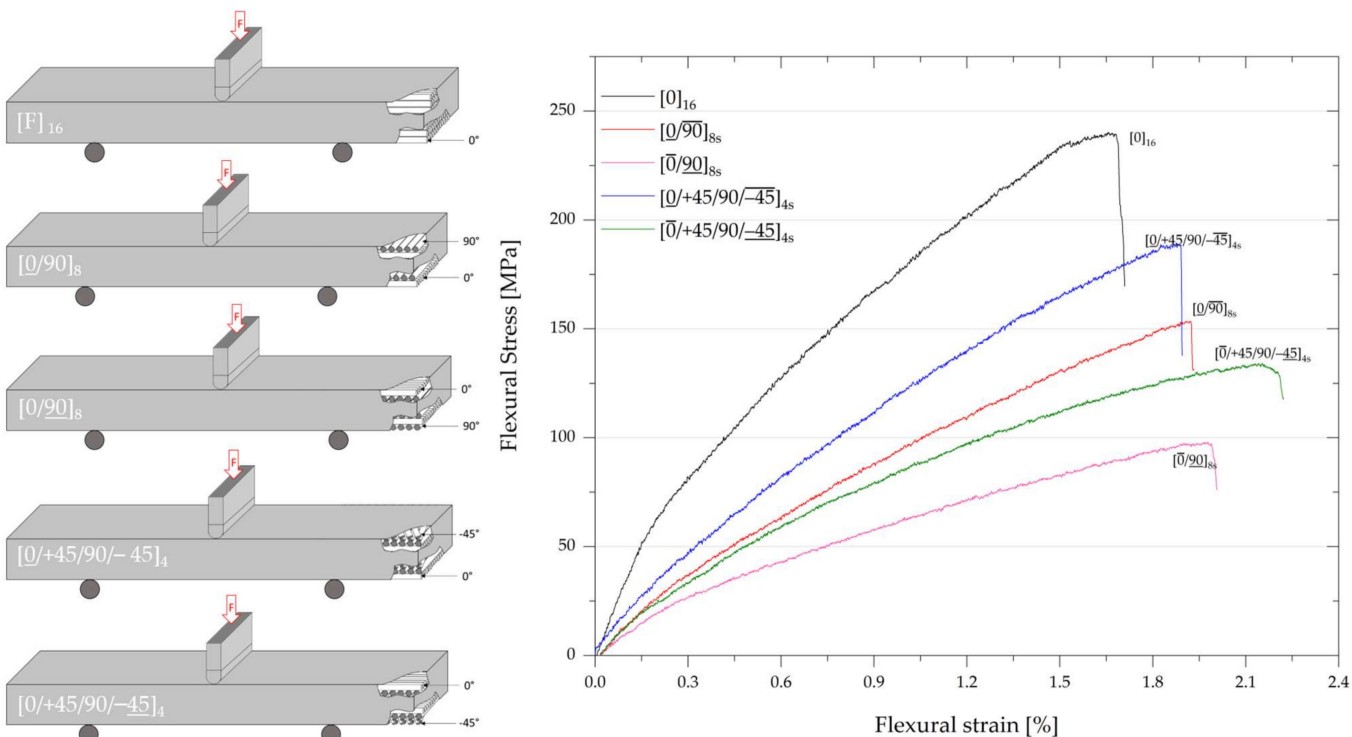

**Figure 1.** Configuration of flexural laminate samples (**left side**) and stress-strain curves obtained from sample testing (**right side**).

**Table 1.** Flexural mechanical properties of studied laminates.

| Laminate Designation | E (GPa) | σ (MPa) |
|---|---|---|
| $[0]_{16}$ | $25.92 \pm 1.14$ | $244.46 \pm 4.33$ |
| $[\underline{0}/90]_8$ | $12.77 \pm 0.98$ | $157.39 \pm 2.52$ |
| $[0/\underline{90}]_8$ | $9.31 \pm 1.29$ | $103 \pm 6.51$ |
| $[\underline{0}/+45/90/-45]_4$ | $15.31 \pm 0.52$ | $192 \pm 1.26$ |
| $[0/+45/90/\underline{-45}]_4$ | $9.65 \pm 0.77$ | $145 \pm 4.82$ |

The second-highest flexural properties are obtained with the $[\underline{0}/+45/90/-45]_4$ laminate samples, which have a flexural modulus and strength of 15 GPa and 192 MPa, respectively. The flexural modulus and strength for $[\underline{0}/90]_8$ laminates are approximately 12 GPa and 157 MPa, respectively. However, by orienting the bottom layers at $\pm 45°$ and $90°$ ($[0/+45/90/\underline{-45}]_4$ and $[0/\underline{90}]_8$), there is a significant decrease in the flexural properties.

The curves in Figure 1 also show that this is the case: the orientation of the bottom layer significantly affects the flexural strain. A substantial increase in the flexural strain accompanied by a decrease in the flexural performance was observed in the case of $[0/+45/90/\underline{-45}]_4$ and $[0/\underline{90}]_8$ laminates.

### 3.2. Predictions of the Damage Patterns in Laminates Using DIC Strain Field Measurements

The full-field strain measurements through DIC are useful for understanding how the stacking sequence affects the laminate's flexural behaviors. Analysis of the full-field strain also permits the detection of mesoscopic intra-laminar cracks in the laminate. The material damage can be discussed using the strain field discontinuity.

The evolution of cracks can be visually correlated to the strain field, as presented in Figures 2–4. Nevertheless, this correlation can only be used if the cracks are close to the outer sample surfaces.

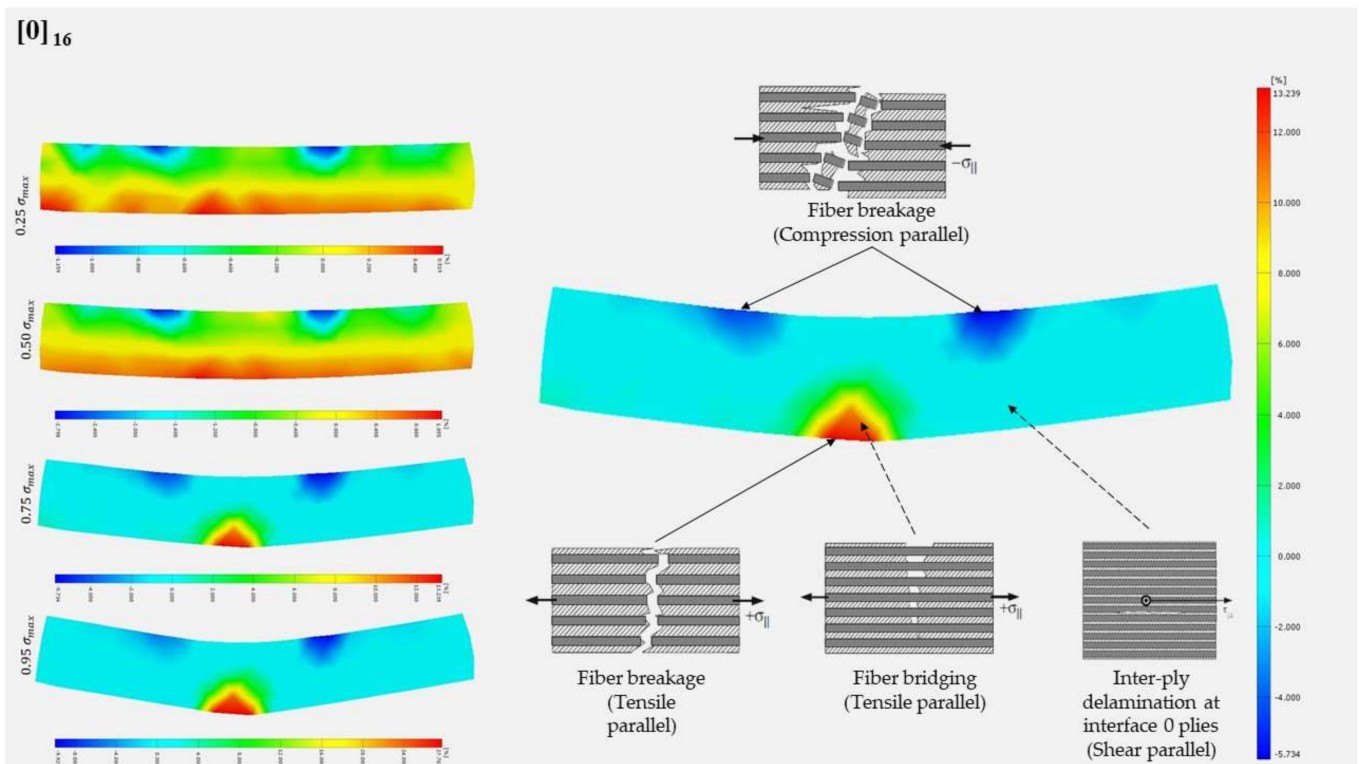

**Figure 2.** $\varepsilon_{xx}$ strain measured by DIC at various load levels for $[0]_{16}$ laminate.

Figures 2–4 present the strain field along the sample length ($\varepsilon_{xx}$). The left side of the figures illustrates strain fields at different load stages (from 25% to 95% of the ultimate flexural load (UFL)). The right side of the figures presents a broader illustration of the strain field at 75% of the UFL. As is known in the flexural load, the upper half of the laminate sample is subjected to a compressive load, and the bottom half of the sample is subjected to a tensile load. Thus, strain fields are illustrated by expected failure mechanisms using a micromechanical representation.

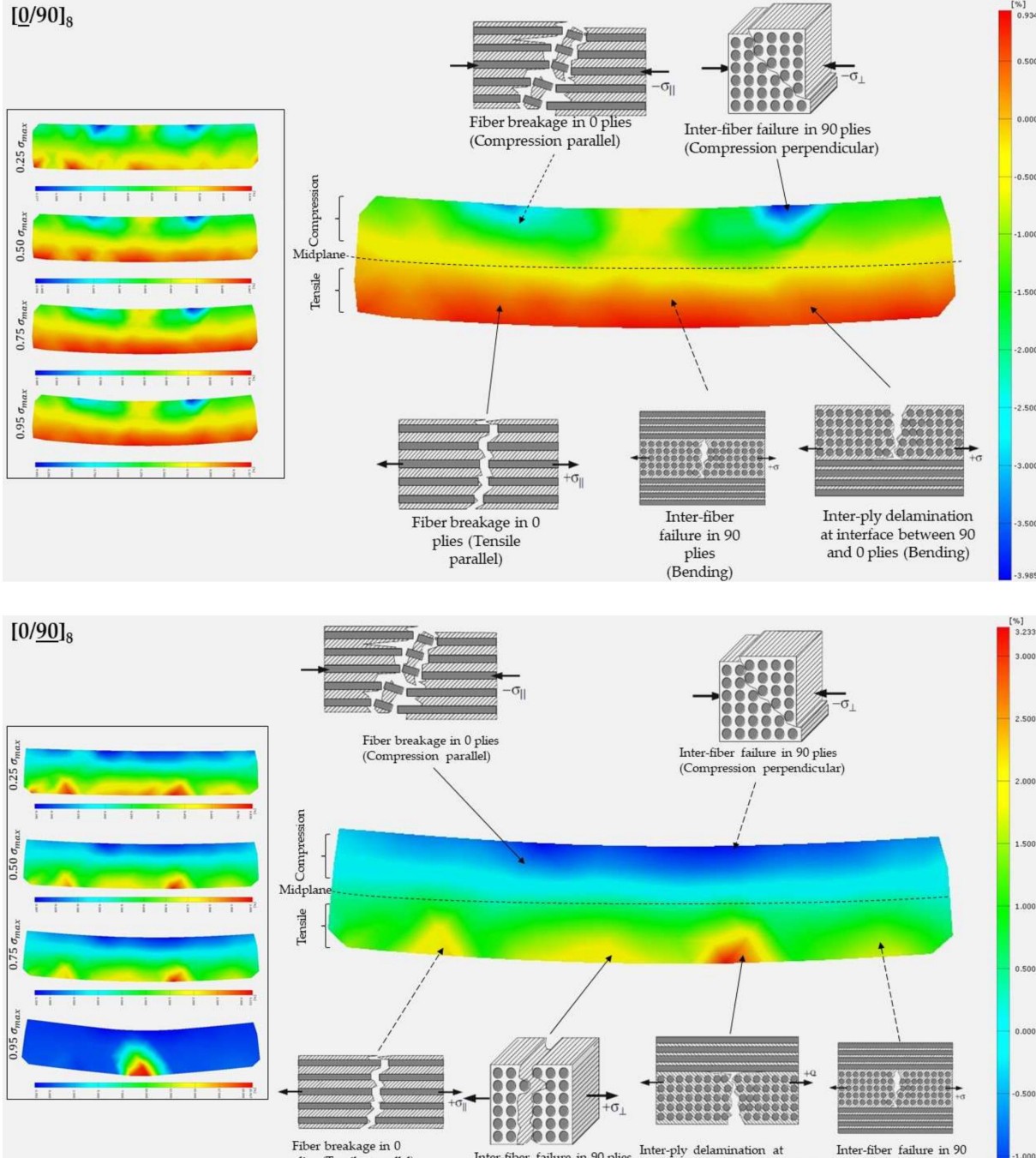

**Figure 3.** DIC measured $\varepsilon_{xx}$ strain at various load levels for $[\underline{0}/90]_8$ and $[0/\underline{90}]_8$ laminates.

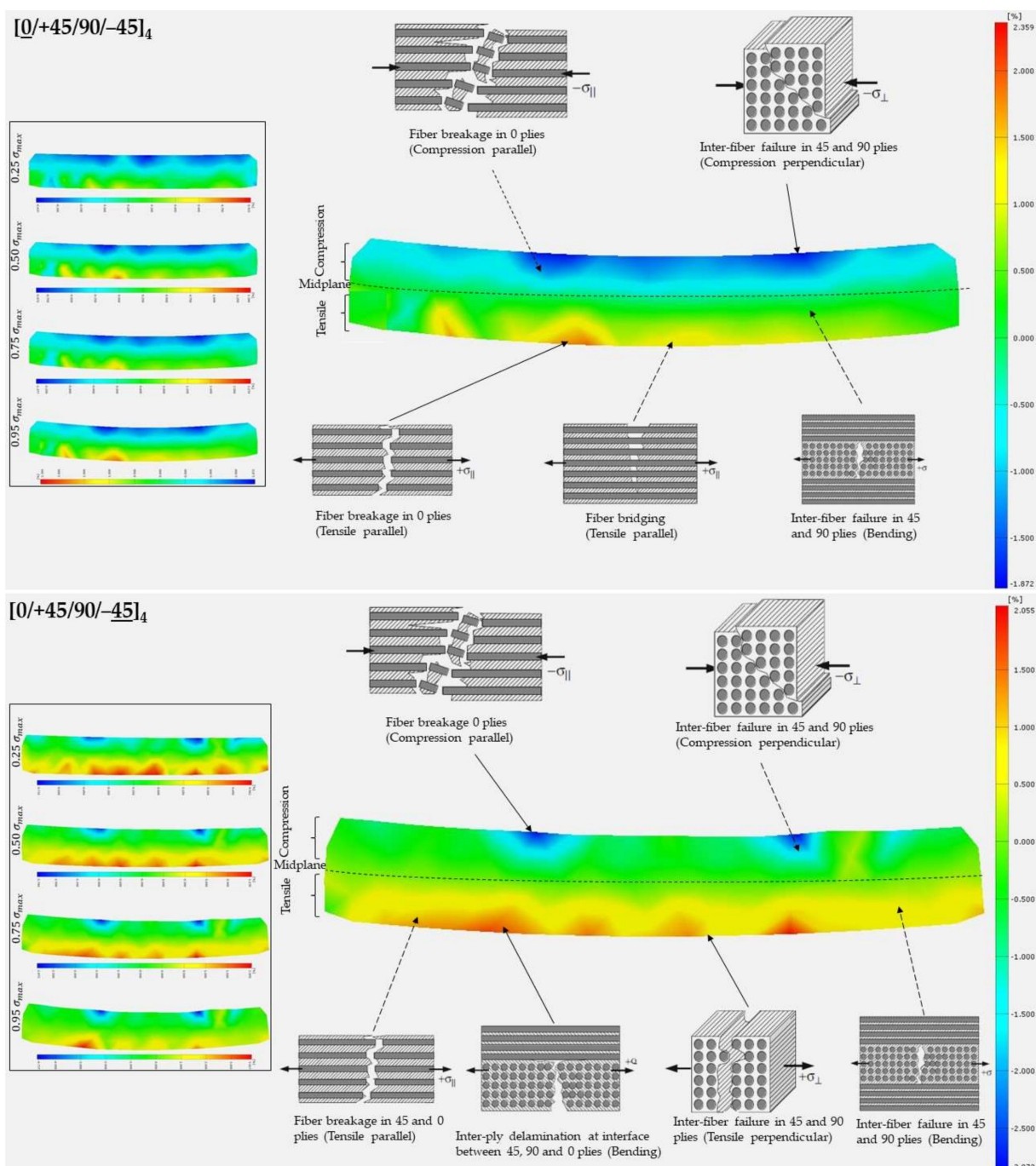

**Figure 4.** DIC measured $\varepsilon_{xx}$ strain at various load levels for ([0/+45/90/−45]$_4$ and [0/+45/90/−45]$_4$) laminates.

### 3.2.1. Analysis of the Strain Fields of Unidirectional Laminate ([0]$_{16}$)

Figure 2 presents the strain field of unidirectional composites [0]$_{16}$. At 25% of the UFL, the strain field reflects a quasi-uniform strain distribution, from a tensile strain of about 0.5% to a compressive strain of about −0.5% throughout the sample thickness. However, a significant compressive strain of about −1.1% was concentrated in two points near the load application position, which indicates damage onset. The load increase to 50% has only doubled the strain values. With further external loading, significant tensile strain is developed at the center (opposite to the load application point). It grows along the loading direction, depicting a major crack leading to the final failure of the composite. This position

is at the center of the growing crack and, thus, of maximum crack opening. Therefore, the similarity between the strain field at 75% and 95% of the UFL is due to the dominant tensile strain concentration in this position. In this case, the compressive strain can be attributed to matrix cracking and compressive fiber breakage. Additionally, tensile strain concentration can be associated with tensile fiber breakage without neglecting a possible fiber bridge and inter-ply delamination at the interface of 0° layers.

### 3.2.2. Analysis of the Strain Fields of Cross-Ply Laminates ([0/90]$_8$)

Figure 3 presents the strain field for the cross-ply stackings ([$\underline{0}$/90]$_8$ and [0/$\underline{90}$]$_8$). In the case of [$\underline{0}$/90]$_8$ laminate, the increase in the external force developed a global strain field similar to that of unidirectional composites [0]$_{16}$ (especially when [0]$_{16}$ is loaded at 50% of the UFL). This trend confirms that the 0° layer mainly governs the global behavior of the laminate.

However, there is a noticeable difference between the strain at the load application position for the [0]$_{16}$ and [$\underline{0}$/90]$_8$ samples. The increased applied force increases the strain value from 0.05% to −0.75% at 25% and 95% of the UFL, respectively. Then, this increase in strain in this position could be related to an inter-fiber failure in the 90° layer at the sample interface and possibly to the compressive fiber breakage of the 0° plies.

Orienting the 90° layer as the bottom layer of the [0/$\underline{90}$]$_8$ samples significantly affects the strain fields under flexural load. Zero strain values starting at the midplane of the sample are gradually moved to compressive strain (on the upper half of the sample), with a more homogeneous distribution through the thickness and higher strain rates with increasing load. However, the tensile strain on the bottom half of the sample seems to be significantly influenced by inter-fiber failure. The increase in load levels induces a strain concentration at the crack growth positions on the edge of the sample. Inter-fiber failure results from forming cracks in the matrix region between the 90° oriented fibers and along the interface between fiber and matrix. Herein, inter-fiber failure is assumed to initiate the final failure crack at the center position of the sample. The strain field at 95% of the UFL confirms this statement, with a concentrated strain of about 20% (Figure 3).

### 3.2.3. Analysis of the Strain Fields of Quasi-Isotropic Laminates ([0/+45/90/−45]$_4$)

Figure 4 illustrates the effect of ±45° layers on the strain fields of the studied laminates. For [$\underline{0}$/+45/90/−45]$_4$ laminate, the increase in the load levels shows an expansion of the compressive strain values from about −0.45% to −1.9% at 25% and 95% of the UFL, respectively. The compressive strain distribution depicts a broad distribution of cracks, where ±45° layers are sensible to develop compressive inter-fiber failure.

A widespread crack distribution is expected in the tensioned layers (as observed with [0]$_{16}$ laminate) for unidirectional fiber-reinforced material. However, the tensile strain field shows that increasing the load mainly induces a strain concentration in a particular position at the bottom surface of the sample. The strain range of Figure 3 indicates the extension of the crack in red areas, while green colors resemble areas where the strain is still less than 0.5%. Subsequently, reducing the total number of 0° layers causes and favors crack initiation at the bottom layer. In the far-field of ±45° and 90° layers, inter-fiber failure may occur on the length scale of a few fiber diameters. Still, it may also span the entire thickness of a ply and propagate onto adjacent layers in the form of inter-ply delamination at the interface between 90° and +45° layers.

In the case of the [0/+45/90/$\underline{-45}$]$_4$ laminate, the upper half of the fracture plane subjected to a dominant compressive normal load shows similar strain fields to the [0]$_{16}$ laminate at 25% of the UFL. Herein, the concentrated compressive strain of about −4% at laminate failure in two distinct points (near the load application position) can be attributed to the compressive breakage of the unidirectional fiber and inter-fiber failure in ±45° and 90° angled layers. The lower part of the fracture plane subjected to a normal tensile load shows triangular red areas representing strain concentration near the −45° bottom layer. Subsequently, the increase in load levels increases the compressive strain value to about

4% at laminate failure. This is caused by the substantial inter-fiber failure in ±45° and 90° layers. After crack growth, inter-layer delamination at the interface between 45, 90, and 0 layers spreads out and propagates along the free surface and into the volume of the laminate.

### 3.3. Flexural Damage Monitoring Using Acoustic Emission

As the first step towards a better understanding of how stacking sequence affects the laminate behaviors, the relationship between the applied force, the detected acoustic emission signals as a source of damage, and the liberated acoustic energy shall be investigated. To this end, Figure 5 presents this relationship for the different laminates studied.

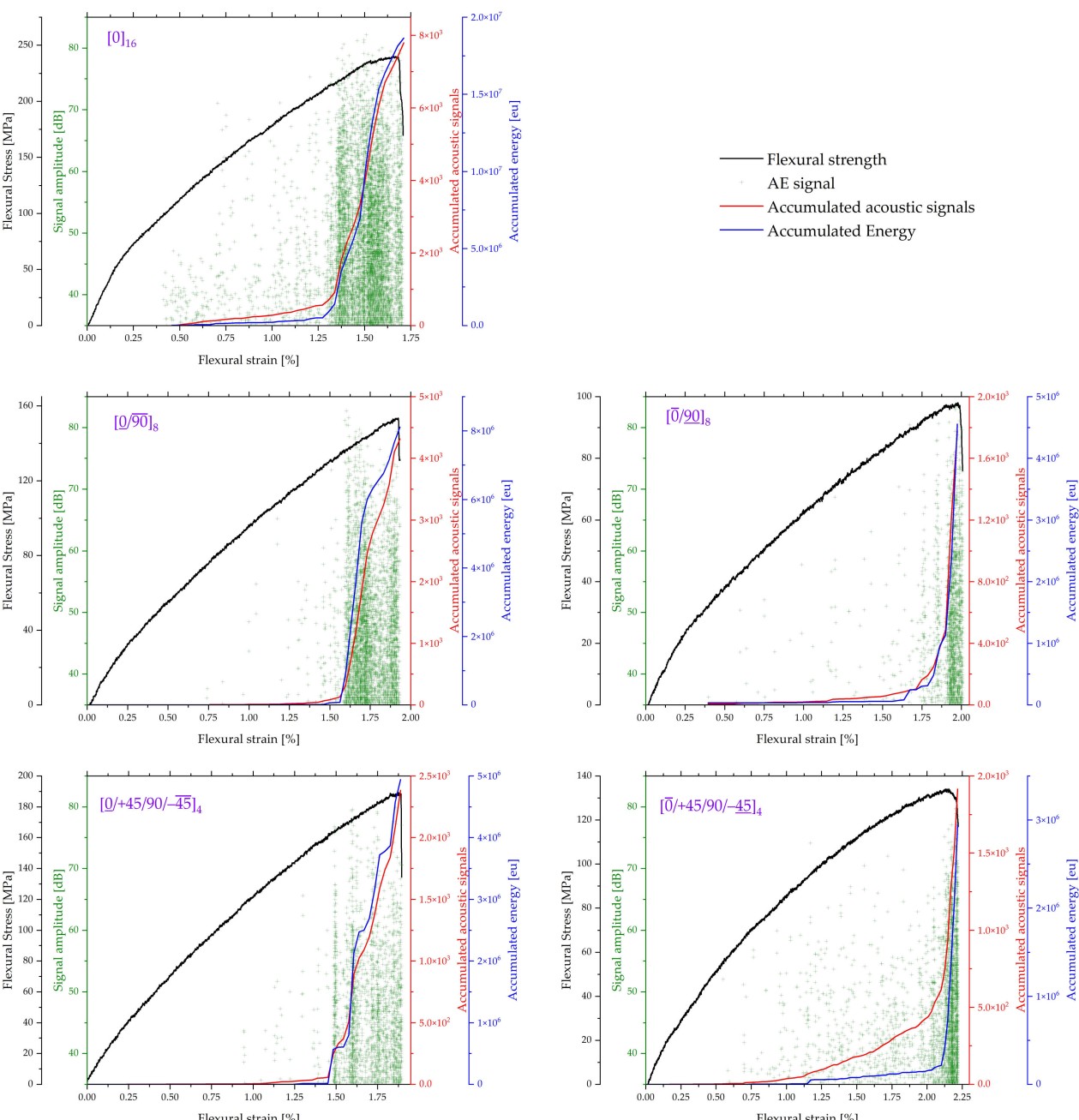

**Figure 5.** Relationship between the applied force, the detected acoustic emission signal amplitude, and the accumulated acoustic energy.



To allow more general conclusions on detected AE signals related to the occurrence of damage, the results of Figure 5 may be discussed with respect to strain levels, the signal amplitudes of each detected event, and the evolution of the accumulated AE energy curves.

### 3.3.1. Global Damage Behavior of Unidirectional Laminate ($[0]_{16}$)

For the $[0]_{16}$ laminate, the acoustic emission activity started at strain levels less than 0.5% while the signal amplitude was still within the 35–50 dB range. The occurrence of high amplitude signals for a strain level between 0.7% and 1.3% indicates that the detected acoustic emission signals could be assigned mainly to the occurrence of new damage types.

Remarkably, the accumulated acoustic energy represents only about 4% of the total energy. With increased load, acoustic emission signals tend to be higher in amplitude and number. This can be partially attributed to the development of previous failure mechanisms and the formation of new ones. From the viewpoint of contributions to the final failure of the laminate, the significant increase in the number of signals and resulting acoustic energy is caused by the occurrence of 95% of damage. Thus, critical damage is expected to initiate at 75% of the ultimate flexural strain.

In this case, the initiation of acoustic activity depicts the onset of matrix cracking. Since the distinction between interfacial failure and matrix cracking intensification is hardly justified, the increase in the number of signals at a strain level between 0.7% and 1.3% could be first assigned to both of them. Therefore, the distinction between the onset of acoustic emission signals and fiber breakage cannot be identified from the results of Figure 5. However, the higher energy release at 75% load to failure is likely due to their development.

As previously reported [2], in the case of unidirectional fiber, the average onset of matrix cracking occurs at load levels between 5 and 30% of the ultimate load. In comparison, fiber breakage signals are quite likely to initiate at around 54% of the ultimate failure load.

### 3.3.2. Global Damage Behavior of Cross-Ply Laminates ($[0/90]_8$)

Damage behaviors of the cross-ply laminates are evaluated using the recorded acoustic activity in both $[\underline{0}/90]_8$ and $[0/\underline{90}]_8$ laminates, as described in Figure 5.

Compared to the unidirectional laminate, acoustic emission signals for $[\underline{0}/90]_8$ laminate were detected at a higher strain level of about 0.75%. The absolute number of signals was found to be negligible until a sudden rise at 1.6%. In this case, 97% of the damage occurs after 82% of the ultimate strain. As discussed before, the distinction between the failure mechanisms associated with acoustic emission signals cannot be achieved. However, the orientation of the fiber layers at 90° generally tends to increase the contribution of matrix cracking to the final failure of the laminate. In addition, the contribution of interfacial failure must increase due to the expected initiation of delamination at the interface between the 0° and 90° layers due to the propagation of the matrix cracking.

Investigations of the effect of the bottom layer orientation on the global damage behavior of the laminates could be achieved using the recorded acoustic activity for the $[0/\underline{90}]_8$ laminate of Figure 5. In this case, acoustic emission signals were detected at a lower strain level and close to the strain value of the matrix cracking onset of the $[0]_{16}$ laminate. Even if the absolute number of signals were found to increase to 4% of the total amount with increasing strain to 1.6%, the accumulated energy allows attributing them to matrix cracking. Notably, the first important increase in acoustic activity was observed at a strain range between 1.6% and 1.78% (10% of the acoustic signals and 7% of the total accumulated energy). This change is thus attributed to damage occurring after the onset of matrix cracking. A second notable increase in acoustic activity was observed at a strain range between 1.78% and 1.89%, where 17% of the acoustic signals and 18% of the total accumulated energy were recorded in this strain range. However, the major amount of acoustic signals (72%) is detected at a distinct strain level (95% of the ultimate failure).

Compared to unidirectional laminates, the systematic decrease in the laminate strength for the cross-ply laminates is expected to yield different damage behavior. The quantified

released energy was one order of magnitude lower in the case of $[\underline{0}/90]_8$ laminate, where the total number of acoustic signals was dropped to half.

As previously demonstrated, acoustic activity is lowest for $[0/\underline{90}]_8$ laminate. The corresponding accumulated energy and number of signals were two times lower than those of $[\underline{0}/90]_8$. Looking more precisely at the first increase in the acoustic activity (strain between 1.6% and 1.78%), the rise in the number of acoustic signals could be attributed to an increase in the matrix cracking signals in the form of inter-fiber failure in the 90° layer. The second, more significant amount of acoustic signals and released energy (observed at a strain range between 1.78% and 1.89%) could signal delamination initiation at the interface between the 0° and 90° layers. Delamination is due to the development of inter-fiber failure across the 90°layer thickness, which reaches the interface of the adjacent 0° layer.

Finally, the distinct quantities of liberated energy, starting at a strain of about 1.56% for $[\underline{0}/90]_8$ laminate and 1.9% for $[0/\underline{90}]_8$ laminate, seem to be related to a possible increase in the fiber failure signals.

### 3.3.3. Global Damage Behavior of $[0/+45/90/-45]_4$ Laminates

Damage behaviors of the $[0/+45/90/-45]_4$ laminates are evaluated using the recorded acoustic activity of Figure 5. In both diagrams, a lower density of acoustic activity is observed. Such a trend is expected to be proportional to reducing the number of 0° layers in the laminate. Compared to the $[0]_{16}$, the number of signals is reduced by a factor of 3 for the $[\underline{0}/+45/90/-45]_4$ stacking sequence and by a factor of 4 for the $[0/+45/90/\underline{-45}]_4$.

Compared to previous results, the released acoustic energy in the case of $[0/+45/90/-45]_4$ laminate changes into a unique step-like shape. Mainly, four distinct steps are observed, and they originate from a concentrated shift of the acoustic emission activity close to four distinct strain rates of 1.5%, 1.6%, 1.75%, and 1.85%. Therefore, the observed steps are attributed to damage occurrence at four different layer orientations. Each step represents a damage mechanism's onset and/or propagation in one representative layer of the stacking sequence. Herein, and especially for acoustic signals appearing after 1.45% of strain (97% of the total number of signals), the possible damage mechanism distinction seems unrealizable.

A good agreement is found between the previous interpretation of the effect of orienting the bottom layer at 90° (in the case of $[0/\underline{90}]_8$ laminate) and the global damage behavior of $[0/+45/90/\underline{-45}]_4$ laminate (when the bottom layer is oriented at −45°). Overall, the mean strain for critical damage onset is systematically lower than the strain values for the $[\underline{0}/+45/90/-45]$ stacking sequence. Although the detected acoustic signals for a strain level between 0.7% and 2% are found to increase linearly, the released energy remains below the threshold of 5%. Similar to the case of $[0/\underline{90}]_8$ laminate, these might be caused by signals originating from matrix cracking and interfacial failure, where most of them belong to a short range of signal amplitudes between 35 and 55 dB.

Due to the possible interaction of several damage mechanisms and their sharp extent at a short strain range of 4% before the ultimate strain at failure, no further distinction can be made between signal types. Consequently, the final increase in the accumulated energy cannot be attributed to the onset of the fiber failure signals but rather to a combination of the three different damage mechanisms.

### 3.4. Classification of Acoustic Emission Signals and Damage Types

#### 3.4.1. Classification Procedure

In the context of acoustic emission signal classification and assignment to failure mechanisms, various classification methods, ranging from simple approaches to pattern recognition methods, allow the establishment of decisive boundaries to group signals in clusters based on their similarity [25]. In the current investigation, the first step before classification consists of determining the optimal number of clusters using statistical data analysis. The Davies-Bouldin index was used to obtain the optimal number of clus-

ters [25]. The second step consists of signal classification using the k-means algorithm. The classification procedure used in this section is detailed in previous work [2,25].

The assignment of the signal classes to corresponding failure mechanisms yields three signal classes for all specimens investigated. Thus, all the damage mechanisms presented in Figures 2–4 using the micromechanical presentation and various microscopic damage mechanisms discussed previously should be regrouped and assigned to the occurrence of matrix cracking, interfacial failure, and fiber breakage.

The assignment of each signal to corresponding failure mechanisms is carried out and shown in Figure 6. Therefore, the obtained clusters and their evolutions allow a better understanding of how stacking sequence affects the damage behaviors of the laminate based on the relationship between the applied force-strain and the detected failure mechanisms. The stress and strain values for the onset of the three failure mechanisms were obtained from the classified signals to evaluate this relationship quantitatively.

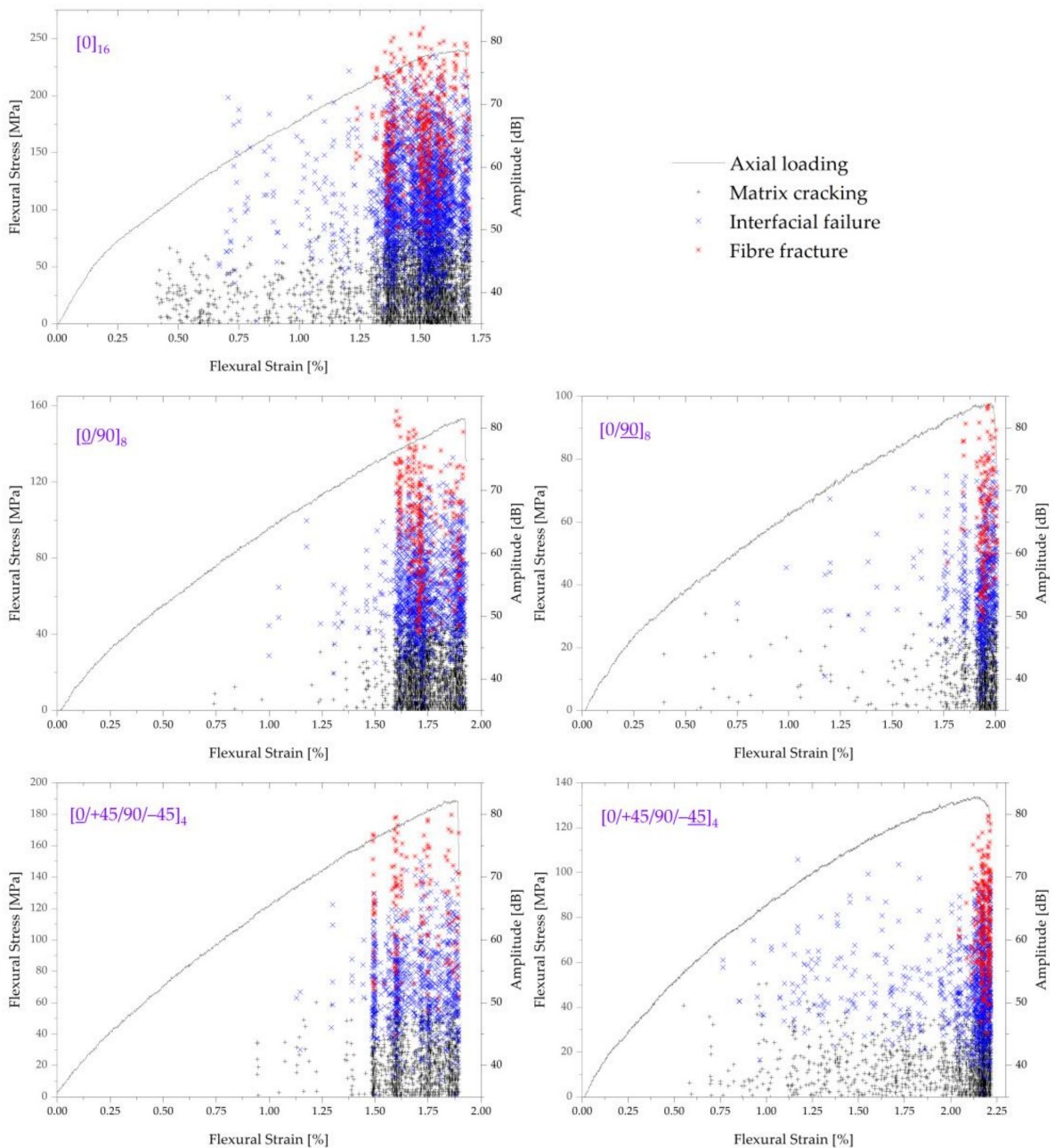

**Figure 6.** Results of the K-means classification of the acoustic emission signals.

### 3.4.2. Detection of Damage Mechanisms Onset and Global Quantification of Their Contribution to the Laminates Failure

Figure 7 presents the mean ratio of the load level at the onset of each damage mechanism and the ultimate failure load for different stacking sequences. The mean values and standard deviations are calculated based on the results of three samples of each laminate. Additionally, accumulated energy was derived from the classified signals and used to calculate each damage mechanism's index of damage contribution (DCi) [2,25]. Accordingly, data sets are labeled by the corresponding equivalent percentage of acoustic signals and the DCi values (values between brackets) for each mechanism in Figure 7.

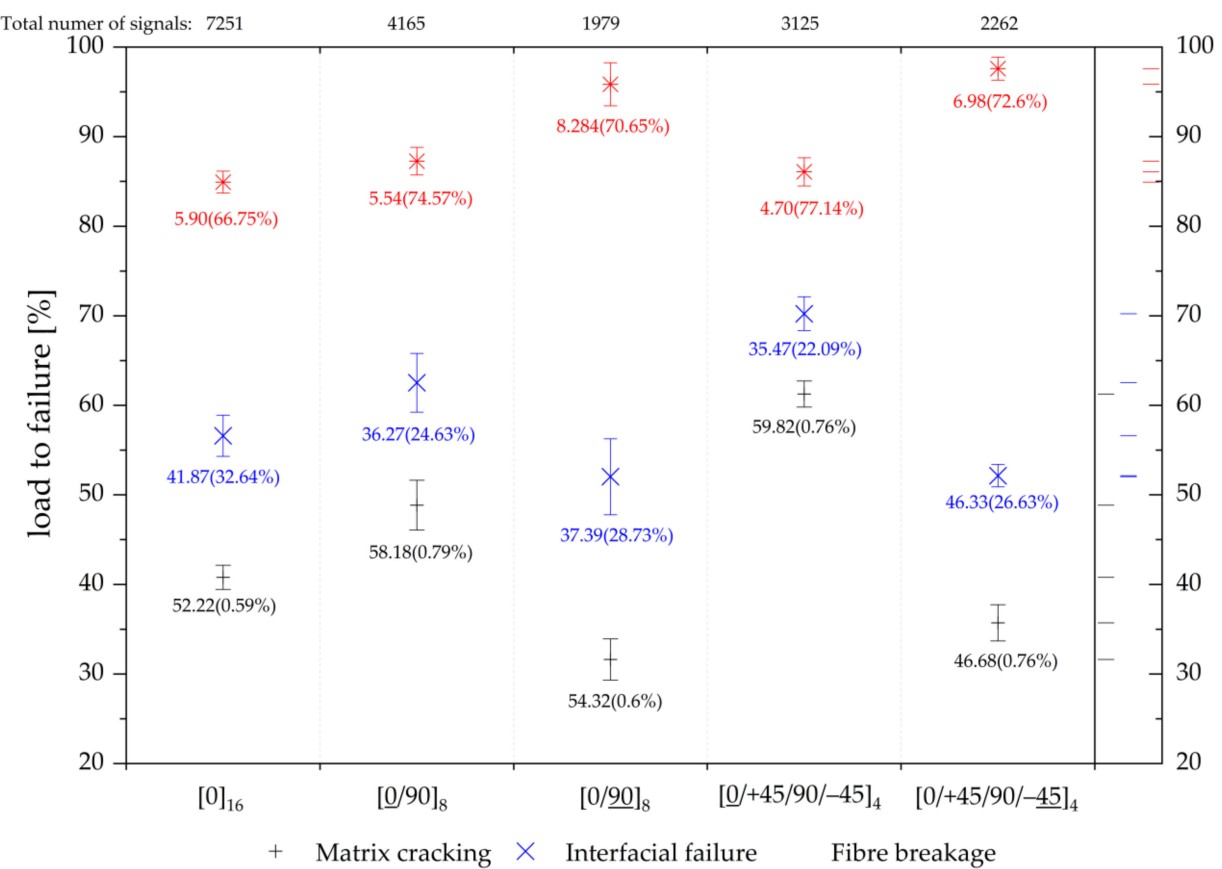

**Figure 7.** The load-to-failure values of the onset of each damage mechanism.

From the results of Figure 7, the average onset of matrix cracking is found at load levels between 30 and 60% of the UFL. Although stacking sequences seem to have a significant effect, where the load ratio is seen to increase in the cases of [0/90] and [0/+45/90/−45] compared to the [0] stacking sequence, orienting the bottom layers at 90° and −45° induces a premature onset of matrix cracking.

Regarding the effect of stacking sequences and bottom layer orientation, the progress of interfacial failure damage is similar to matrix cracking. Herein, the average load for the onset of interfacial failure is located between 50 and 70% of the UFL. While the bottom layers are oriented at 0°, the average onset of fiber breakage is quantified at around 85% of the ultimate failure load for all laminates. However, orienting the bottom layers at 90° and −45° delayed the beginning of fiber breakage and development in 95% of the UFL.

Figure 7 reveals some interesting details by evaluating the individual failure mechanisms' contributions. Here, the relative contribution was quantified according to the number of acoustic signals of each damage mechanism and their contribution to the overall laminate failure.

A systematic decrease in the total number of signals for each laminate was observed, where the [0/+45/90/$-$45] and [0/$\underline{90}$] laminates are characterized by lower acoustic activity. As a first conclusion, matrix cracking signals represent the most important portion of recorded signals for all studied laminates (ranging between 46.68% and 59.82% of the total number of signals). Despite this high acoustic activity, their calculated damage contribution index (DCi) from the absolute released energy was very low or even negligible (<1% of the total accumulated energy).

However, it is clearly shown that the interfacial damage mechanism makes a significant contribution. Although moderate acoustic activity ranges between 35.47% and 46.33%, this damage mechanism contributes 22.09% to 32.64% of the induced damages for all studied laminates.

As is typical for composite materials, fiber failure releases the highest acoustic energy. Even when started close to the ultimate load to failure, representing less than 8.28% of the total acoustic activity, their DCis show major contributions with a minimum of 66.75%.

The relationship between the acoustic signal type and the amount of released energy was discussed in recent research and correlated to the signal waveforms [26–29]. The lowest energy release by the matrix cracking signals was attributed to their low amplitude and slow change with time. In contrast, higher amplitude and multiple peaks characterize the waveform of interfacial damage, where the resulting energy is much higher than matrix cracking. Finally, fiber failure signals occur suddenly, have a shorter rise time, and have a larger amplitude when the energy rapidly ranges at a high level.

3.4.3. Decomposition of the Acoustic Emission Signals into Different Microscopic Failure Mechanisms

Fundamentally, the detected acoustic emission signals could represent various microscopic damage mechanisms. As shown by the micromechanical schematic illustration of possible damage types in Figures 2–4, flexural loading induced crack initiation and development at the interface between fiber and matrix or between reinforcement layers. Additionally, it caused fiber failure by a tensile or compressive load and complex interactions and growth scenarios of the different damage mechanisms. The matrix area surrounding this fiber must be fully cracked to initiate fiber failure. In addition, a void or inter-fiber crack acts as a stress concentrator at the fiber's surface. Another issue encountered in fiber-reinforced laminates is the presence of an inter-fiber crack in off-axis plies relative to the load axis, which causes the initiation of inter-ply delamination.

The significance of the acoustic emission analysis can be substantially increased by the distinction between the different failure types mentioned below. However, due to the complexity of establishing subclusters, especially for matrix and interfacial damage, it was found to be more reasonable, given the reports in the literature [11,30] and the typically expected failure mechanisms for fiber-reinforced polymers, to form three representative clusters. In the following, the results of Figures 4–7 are evaluated and correlated to get a further decomposition of the three main clusters into different microscopic failure mechanisms and to discuss their potential interactions.

Decomposition of the Acoustic Emission Clusters for Unidirectional Laminate ([0]$_{16}$)

For the [0]$_{16}$ stacking sequence, matrix cracking has the most important acoustic activity. However, interfacial failure represents 41.87% of the acoustic activity and contributes 32.64% to laminate failure. Remember that the strain field (of Figure 2) before reaching 50% of the UFL of this laminate was found to have a quasi-uniform strain distribution. Additionally, the classified acoustic signals in Figure 6 have shown a distinct onset of matrix cracking and interfacial damage. Here, the first damage onset is typically given by the initiation of microcracks along the fiber axes, which explain the quasi-uniform tensile strain distribution of the CCD strain field. This was followed by a low rate of interfacial failure activity. Hence, these signals are usually attributed to the development of microcracks on the interface between fiber and matrix. However, the onset of fiber failure was seen as

the initiator of a growing damage state and can be directly related to the highly increased rate of matrix cracking and interfacial failure mechanisms. Here, trend monitoring of their acoustic activities could indicate the onset of inter-layer delamination at the interface of 0° plies. Specifically for this case, the three mechanisms show a similar trend regarding their intensification by increasing the applied load and a large strain area (from 1.3% to 1.7%). It can be concluded that the relevant failure progression is given by the first onset of matrix cracking, followed by the failure of fibers at the edge of the specimen. Subsequently, the increased applied force seems to propagate the above damage through the specimen thickness, initiating inter-layer delamination.

Decomposition of the Acoustic Emission Clusters for Cross-Ply Laminates ([0/90]$_8$)

In the case of [$\underline{0}$/90]$_8$ laminate, the strain field measurement has shown a global strain distribution governed by the 0° plies (Figure 3). However, Figure 7 shows a significant increase in fiber failure, contributing to the laminate failure by a DCi of 74.57%. In general, the diminution of the 0° plies must induce a decrease in fiber failure signals and, consequently, a reduction in their absolute liberated energy. Here, the total number of acoustic signals was decreased without decreasing the proportion of released energy due to fiber failure. In addition, Figure 7 shows an apparent decrease in interfacial failure signals and their derived contribution to failure to 24.63%, compared to 32.64% for unidirectional laminate. Combined with the earlier onset of interfacial damage, as seen in Figure 6, this evolution of the index of damage contribution of both fiber and interfacial failure mechanisms may reflect a transformation of the crack onset and development compared to those observed with the [0]$_{16}$. Since the fiber orientation significantly affects the strain and stress states of the individual layers, interfiber cracks can occur either in parallel with or transverse to the load axis. As long as the lower part of the sample is subject to a normal tensile load, the onset of matrix cracking signals could be attributed to an inter-fiber failure in 90° oriented layers. Depending on the applied force, the 0° oriented layers typically act as a crack stopper. However, such inter-fiber cracks can induce high-stress concentration at the crack tip and become a typical initiator of fiber failure and inter-layer delamination propagating at the interface between 90° and 0° adjacent layers.

Moreover, the upper half of the fracture plane is subjected to a dominant compressive normal load. Thus, a portion of the recorded acoustic activity is attributed to damage onset and development on the upper half of the sample. Therefore, matrix cracking signals could arise from inter-fiber failure, and fiber failure signals can originate from fiber crushing due to compressive loading.

In the case of [0/$\underline{90}$]$_8$ laminate, the obtained result of the strain field measurements showed a particular onset and development of inter-fiber failure in the bottom layer of the laminate. Inter-fiber failure results from initiating cracks in the matrix region between the 90° oriented fibers. This concentrated inter-fiber failure has also been confirmed by the analysis of the global damage behavior of [0/$\underline{90}$]$_8$ laminate and the obtained results of signals classification in Figure 6. A concentration of interfacial failure signals was observed at a strain range between 1.78% and 1.89%, where 17% of the acoustic signals and 18% of the total accumulated energy were recorded in this strain range. In this case, matrix cracking signals can be attributed to inter-fiber cracks in the 90° fiber layers. Suppose such laminates are subjected to a normal tensile load. In that case, this type of failure naturally results in significant inter-layer delamination, which explains the concentrated acoustic signal relative to the interfacial failure at strain ranges of 1.78% and 1.89%. Results from Figure 7 for the index of damage contribution support this interpretation. Interfacial failure is characterized by a higher effect on laminate failure of 28.73%, compared to 24.63% in the case of [$\underline{0}$/90]$_8$ laminate.

Inter-fiber cracks can propagate within the matrix or at the interface between fiber and matrix, depending on the applied force. Suppose the applied tensile normal load exceeds the local strength of the interfacial strength between fiber and matrix. In that case, interfacial failure can be further decomposed into fiber–matrix debonding and fiber–matrix pull-out.

Decomposition of the Acoustic Emission Clusters for $[0/+45/90/-45]_4$ Laminates

In the case of $[\underline{0}/+45/90/-45]_4$ laminates, further decomposition of the acoustic signal is more complicated due to the effect of $\pm 45°$ oriented layers. However, DIC strain field measurements showed a tensile strain concentration in a particular position at the bottom surface of the sample. The evolution of the strain field under increased load (Figure 3) indicates the propagation of cracks at the $0°$ bottom layer, which can be initiated to the extent of $\pm 45°$ and $90°$ layers.

In all cases, cracks initiate or propagate within the matrix or at the interface between fiber and matrix. Thus, matrix cracking failure mechanisms can be correlated to inter-fiber cracks.

A particular stair-step shape of the accumulated energy was observed in Figure 5. Four distinct steps were attributed to a concentrated shift of the acoustic emission activity, which was close to four distinct strain rates of 1.5%, 1.6%, 1.75%, and 1.85%. Correlating this interpretation to the results of the classification of signals in Figure 6 provides a better understanding of the failure mechanism sequences and, consequently, a better decomposition.

Even if a very small number of matrix cracking and interfacial failure signals are observed at the onset of acoustic activity, they can cause the inherent initiation and growth of damage, representing the first step of the accumulated energy. Thus, the concentrated damage at the stain of about 1.5% can be attributed to an inter-fiber failure due to the normal tensile load, which is more likely in the $90°$ layer than in $\pm 45°$ layers.

While the acoustic signal between the four distinct strains of 1.5%, 1.6%, 1.75%, and 1.85% reflects a damage progression under applied load, the recorded signals are of matrix cracking and interfacial failure types. Therefore, the resulting damage growth is a mix of crack propagation and significant interfacial failure. However, the index of damage contribution from interfacial failure was found to decrease to 22.09%, compared to the $[\underline{0}/90]_8$ laminate. In particular, the decrease in the DCi was not accompanied by a reduction in the interfacial failure signal proportion (35.47% for $[\underline{0}/+45/90/-45]_4$ laminate and 36.27% for $[\underline{0}/90]_8$ laminate). Knowing that inter-layer delamination signals are of higher released energy may allow concluding that interfacial failure signals are mainly of fiber–matrix debonding and fiber–matrix pull-out types.

In the case of $[0/+45/90/\underline{-45}]_4$ laminate, Figure 5 shows the development of 20% of the total number of damage signals for strain between 0.7% and 2%. As seen from the global damage behavior evaluation, their amount of absolute acoustic energy was limited to 5%. However, the strain field measurements in Figure 4 have shown that at this loading state, these acoustic signals, identified in Figure 5 as matrix microcracking and interfacial failures, generate subcritical damage growth near the $-45°$ bottom layer. Within a short strain range of 4% to the ultimate load to failure, the generated damage has initiated macroscopic crack growth, leading to the final failure of the laminate.

In such a case, both matrix cracking and interfacial failure that generates subcritical damage growth can be attributed at first to inter-fiber cracks, fiber–matrix debonding, and fiber–matrix pull-out, respectively.

As a distinct difference, macroscopic crack growth liberates 95% of the total accumulated energy. Despite the decrease in the total number of acoustic signals, Figure 7 shows a distinct contribution of interfacial failure with a DCi of 26.63% and a higher signal proportion of 46.33%. Subsequently, the combination of these two last findings allows attributing the interfacial failure signals to inter-layer delamination at the interface between $\pm 45°$, $90°$, and $0°$ layers, in addition to fiber–matrix debonding and fiber–matrix pull-out, which spread out and propagate along the free surfaces and into the volume of the laminate.

Finally, for both $[0/+45/90/-45]_4$ and $[0/+45/90/\underline{-45}]_4$ laminates, the upper half of the fracture plane is subjected to a dominant compressive normal load. It is readily understood from the strain field of Figure 4 that craking matrix signals should include compressive inter-fiber failures in addition to tensile inter-fiber cracks. In the same way, fiber failure signals should also include compressive fiber breakage.

## 4. Conclusions

Unidirectional, cross-ply, and quasi-isotropic flax laminates were tested under flexion to highlight the effect of the stacking sequence on their mechanical and damage behaviors. Full strain field measurements and in-situ damage event detection were achieved using DIC and AE, respectively.

In summary:

- Acoustic emission activity starts at less than 0.5% strain for the $[0]_{16}$ laminate, and critical damage is expected to initiate at 75% of the ultimate flexural strain.
- For the $[0/90]_8$ laminate, acoustic emission signals were detected at a higher strain level of about 0.75%. The orientation of the fiber layers at 90° generally tends to increase the contribution of matrix cracking on the final failure of the laminate. Moreover, the contribution of interfacial failure must increase due to the expected initiation of delamination at the interface between the 0° and 90° layers due to the propagation of the matrix cracking.
- The bottom layer orientation affects the global damage behavior of the laminates. Acoustic emission signals were detected at a lower strain level for the $[0/90]_8$ laminate with the bottom layer oriented differently. A major amount of acoustic signals (72%) is detected at a distinct strain level (95% of the ultimate failure).
- The quantified released energy is one order of magnitude lower for the $[0/90]_8$ laminate compared to unidirectional laminates, and the corresponding accumulated energy and number of signals were two times lower than those of the $[0/90]_8$.
- For the $[0/+45/90/-45]_4$ laminates, compared to the $[0]_{16}$, the number of signals is reduced by a factor of 3, and a lower density of acoustic activity is observed.
- The $[0]_{16}$ laminate had the most important acoustic activity due to matrix cracking. Interfacial failure contributed to 41.87% of the acoustic activity and 32.64% to the laminate failure. The failure progression was given by the first onset of matrix cracking, followed by the failure of fibers at the edge of the specimen. The $[0/90]_8$ laminate showed a significant increase in fiber failure, contributing to the laminate failure by 74.57%. The interfacial failure signals decreased to 24.63% compared to 32.64% for the $[0]_{16}$ laminate. Inter-fiber failure in the 90° fiber layers was observed in the bottom layer of the laminate. Interfacial failure was characterized by a higher effect on laminate failure of 28.73%. In the case of $[0/+45/90/-45]_4$ laminates, the strain field measurements showed that failure started with matrix cracking and interfacial failure and was then followed by fiber failure.

**Author Contributions:** Formal analysis, Invastigation, Writing—review & editing, M.H. and Visualization, Supervision, L.L. All authors have read and agreed to the published version of the manuscript.

**Funding:** This research was funded by Natural Sciences and Engineering Research Council (NSERC) grant number CRSNG–RGPIN-2021-02846.

**Data Availability Statement:** The data presented in this study are available on request from the corresponding author.

**Conflicts of Interest:** The authors declare no conflict of interest.

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
