# Peer review of "Combining Digital Image Correlation and Acoustic Emission to Characterize the Flexural Behavior of Flax Biocomposites"

_2673-3161, doi:10.3390/applmech4010021_

Round 1

Reviewer 1 Report

In this research, a combination of digital image correlation (DIC) and acous-

tic emission (AE) is used to locate and classify the type of damage depending on the stacking sequence of the laminate during flexural loading.The article is of scientific and practical interest. I recommend the manuscript for publication

Author Response

Response to Reviewer 1 Comments

Thank you for taking the time to review our manuscript and for your positive feedback. We are pleased to hear that you found our research on using digital image correlation and acoustic emission to locate and classify damage in laminate structures during flexural loading to be of scientific and practical interest.

We appreciate your recommendation for publication. Thank you again for your valuable feedback.

Reviewer 2 Report

The authors combine digital image correlation and acoustic emissions to characterize damage in composites. The content is highly relevant and the paper is recommended for publication after revising the paper based on the following comments:

1) The results shown in figures 2, 3, and 4 uses a color scheme what does not correspond to the same magnitude of strain. Hence comparison of the images is not possible. This has to be corrected and a same colour-scale must be used for all the images that show the strain value distribution. Also the text in the figure that shows the strain values is too small and has to be enlarged. 

2) Also the figures 5 and 6 use different y axis scalings and here also the images cannot be compared. To be comparable the same y axis scale must be used. Also the quality of the images should be improved. 

3) The section 3.4.1 is very brief. It requires more explanation. It is not clear how the dataset was prepared for clustering. Did the DIC data also go into the clustering process ? How was the micromechanical information incorporated into the dataset i.e. labelling ? Was a one-hot encoded scheme used ? Which software package  Eventhough the details might have been discussed in another paper, the specific for of the dataset and the contents of the dataset are of interest for readers. 

4) The conclusions seem to mix the summary and the main findings of the paper. The authors are recommended to provide a bullet point list of the main findings that have resulted from the the investigations presented in this paper. 

Author Response

Response to Reviewer 2 Comments

Thank you for taking the time to review our manuscript and for providing us with constructive feedback. We appreciate your valuable comments and will address each of your points below.

Response to point 1: We acknowledge your concern regarding the color scheme used in Figures 2, 3, and 4. However, the strain values differ between the different materials, and using the same color scale for all the images may result in losing important information. The minimum and maximum strains differ from material to material. Figure 1 has minimum and maximum strain values between -6% and 13%. Figures 2 and 3 have values between 2% and 3%. If we use the same color scale, we must use minimum and maximum strain between -6% and 13%. In this case, Figures 2 and 3 will lose the strain concentration near the zone of interest, and we will lose the significance of the DIC information. Regarding the font size in the figure legends, we would like to draw your attention that the Lavision software used to export DIC results is limited in terms of text size modification. However, the full-resolution images will be available to readers online, which will allow them to zoom in and read the text as needed. Images resolution was enhanced from 300 dpi to 600 dpi.

Response to point 2: We appreciate your comment regarding Figures 5 and 6. The quality of the images was enhanced by adjusting the scale, size, and colors. However, the axis limits can not be equalized due to the measured features’ differences. Unidirectional laminates generate 10 times more energy than others, which dictates the axis scale to represent the data accurately. Otherwise, the reading of figures will be more. Images resolution was enhanced from 300dpi to 600 dpi.

Response to point 3: We understand your concern regarding the brevity of Section 3.4.1. However, in order to lighten the text and as the acoustic emission and the analysis of these data have now become more and more known, we have referred the reader to our previous work, which explains the method used for the classification and answers the questions of the reviewer.

Response to point 4: We appreciate your feedback regarding the conclusions. To address this, we provided a bullet point list summarizing the main findings of our investigation. This will help readers understand our study’s key takeaways more clearly.

Reviewer 3 Report

In this paper, a combination of digital image correlation (DIC) and acoustic emission (AE) is used to locate and classify the type of damage depending on the stacking sequence of the laminate during flexural loading, the paper has many interesting aspects. The research work is of significance because it presents an approach that can be used to characterize the flexural behavior of flax biocomposites.

Author Response

Response to Reviewer 3 Comments

Thank you for taking the time to review our manuscript and for your positive feedback. We are pleased to hear that you found our research on using digital image correlation and acoustic emission to locate and classify damage in laminate structures during flexural loading to be of scientific and practical interest.

We appreciate your recommendation for publication. Thank you again for your valuable feedback.

Reviewer 4 Report

In this paper, a combination of digital image correlation (DIC) and acoustic emission (AE) was used to locate and classify the type of damage depending on the stacking sequence of the laminate during flexural loading. It can be accepted after revision.

1.      The conclusion part needs to be condensed to reflect some quantitative conclusions.

2.      The introduction part needs to be refined to reflect innovation.

3.      In the introduction part, it is suggested to add the application status of DIC and AE in composite materials.

Author Response

Response to Reviewer 1 Comments

Thank you for reviewing our manuscript and for your valuable feedback. We appreciate your positive comments and your suggestions for improvement. We will address each of your points below.

Response to point 1: We appreciate your comment regarding the conclusion section of our paper. We have revised it to be more concise and highlight the quantitative conclusions drawn from our study.

Response to point 2: We understand your concern about the introduction section and agree that it needs to be refined to reflect the innovation of our research. We have revised the introduction to highlight the innovative aspects of our research and the contributions it makes to the field.

Response to point 3: Thank you for suggesting that we add the application status of DIC and AE in composite materials in the introduction. We have updated the introduction to include a brief overview of this field’s current state of the art.